# Modeling the Dynamics of Children’s Musculoskeletal Fitness

**DOI:** 10.3390/ijerph20042938

**Published:** 2023-02-08

**Authors:** Ana Reyes, Raquel Chaves, Olga Vasconcelos, Sara Pereira, Go Tani, David Stodden, Donald Hedeker, José Maia, Adam Baxter-Jones

**Affiliations:** 1Instituto Superior Manuel Teixeira Gomes (ISMAT), 8500-724 Portimão, Portugal; 2Centre of Research, Education, Innovation and Intervention in Sport (CIFI2D), Faculty of Sport, University of Porto, 4099-002 Porto, Portugal; 3Academic Department of Physical Education, Federal University of Technology of Paraná, Curitiba 80230-901, Brazil; 4Centro de Investigação em Desporto, Educação Física, Exercício e Saúde (CIDEFES), Lusófona University, 1749-024 Lisboa, Portugal; 5School of Physical Education and Sport, University of São Paulo, São Paulo 13560-970, Brazil; 6Physical Education, University of South Carolina, Columbia, SC 29208, USA; 7Department of Public Health Sciences, University of Chicago, Chicago, IL 60637, USA; 8College of Kinesiology, University of Saskatchewan, Saskatoon, SK S7N 5B5, Canada

**Keywords:** child, physical fitness, longitudinal, multilevel analysis

## Abstract

This study models children’s musculoskeletal fitness (MSF) developmental trajectories and identifies individual differences related to effects of time-invariant, as well as time-varying covariates. A total of 348 Portuguese children (177 girls) from six age cohorts were followed for three years. MSF tests (handgrip strength, standing long jump and shuttle run), age, body mass index (BMI), socioeconomic status (SES), gross motor coordination (GMC) and physical activity (PA) were assessed. Data were analyzed using multilevel models. Between 5 and 11 years of age, boys outperformed girls in all three MSF tests (*p* < 0.05). Birth weight was positively associated with shuttle run performance (*β* = −0.18 ± 0.09, *p* < 0.05). BMI was positively associated with handgrip strength (*β* = 0.35 ± 0.04, *p* < 0.001) and shuttle run performance (*β* = 0.06 ± 0.01, *p* < 0.001), but negatively associated with standing long jump performance (*β* = −0.93 ± 0.23, *p* < 0.001). GMC was positively associated (*p* < 0.001) with all three MSF tests, while PA was associated with standing long jump (*β* = 0.08 ± 0.02, *p* < 0.05) and shuttle run (*β* = −0.003 ± 0.002, *p* < 0.05) performance only. No school environment effects were found, and SES was not related to any MSF tests. Children’s MSF development showed a curvilinear shape with increasing age, with boys outperforming girls. Weight status and physical behavior characteristics predicted MSF development, while environmental variables did not. Examining potential longitudinal predictors of MSF across multiple dimensions is important to gain a more holistic understanding of children’s physical development as well as to future interventions.

## 1. Introduction

Musculoskeletal fitness (MSF), a multidimensional construct comprising muscle strength, muscle endurance, and muscle power, is an important marker of health status [1]. Differences in children’s MSF are explained by a variety of genetic and environmental factors [2]. Cross-sectional studies of children’s MSF have shown positive associations with factors such as physical activity (PA), birth weight, motor competence, gross motor coordination (GMC), and socioeconomic status (SES) [3,4]. School environments also influence MSF [5]. However, few studies have investigated how all these factors combined impact MSF development.

Longitudinal studies of human growth indicate the pattern of curvilinear growth is consistent, but the timing and magnitude of growth is highly individualized [6,7]. This is also true for MSF outcomes [2]. To adequately distinguish the independent effects of sex, birth weight, SES, school environment, body mass index, GMC, and PA on MSF development, time-invariant and time-varying factors are required; this can only be achieved using longitudinal data and appropriate statistical analysis. This approach accounts for the wide variation shown amongst children’s growth parameters at any given age, and in the velocity of these parameters across age, and requires hierarchical (or multilevel) modeling [8]. 

A longitudinal study by Ruedl et al. [9] reported that physical fitness development in overweight boys and girls was lower than their non-overweight peers. In addition, a study by Rodrigues et al. [10], describing different velocities, found that children with low or average rates of change in MSF and motor competence were more likely to become overweight or obese by the end of childhood, regardless of their sex or initial body mass index (BMI). Also, Haugen and Johansen [11] found that the initially high gross motor competence-grouped children performed significantly better on a multidimensional physical fitness test, compared to the initially low gross motor competence-grouped children, at all time-points. Additionally, Bai et al. [12] reported that higher SES and school size were associated with better MSF profiles. One limitation of these studies is that they did not consider other potential MSF correlates, nor did they consider the hierarchical data structure [9] of the school context [10], or the time dependent effects of PA or GMC [12].

The growth, motor development and cognition study (GMDC) [11] was initiated to address these limitations. The objectives of this paper were: (i) to model intra-individual MSF developmental trajectories, (ii) identify the magnitude of inter-individual differences and their dependence on school contexts, and (iii) investigate the effects of time-invariant (e.g., sex, birth weight and SES) and time-varying (e.g., age, BMI, GMC and PA) covariates on children’s MSF development. We hypothesized that MFS development would show a curvilinear trend (positive for hand grip strength and for standing long jump, but negative for shuttle run), and the trajectories would be greater in boys. Furthermore, we hypothesized that school contexts would influence trajectories, and that time-variant and invariant predictors would have independent significant effects on MFS development.

## 2. Materials and Methods

The GMDC was carried out in Vouzela, a central region of Portugal, between 2013 and 2016. The mixed-longitudinal design has been described in detail elsewhere [13]. In brief, data were collected in six age cohorts, followed for three consecutive years (4–6, 5–7, 6–8, 7–9, 8–10 and 9–11 years). All children from the 19 schools of the region were invited (participation rate ~90%). Children with special needs (*n* = 24) were excluded from the analysis as were 4-year-old’s (*n* = 88) whose GMC was not assessed. Written informed consent was obtained from parents or legal guardians, and the project was approved by local authorities and the ethics committee of the Faculty of Sport, University of Porto (CEFADE 01.2016).

Data from all cohorts (*n* = 348, 177 girls) with complete data were used in the analysis (Table 1). In addition, differences in a set of variables (sex, age, PF tests, birth weight, BMI, GMC, PA, and SES) were analyzed to identify putative differences between those with complete and those with missing data (*n* = 150, 75 girls). Significant differences were found in BMI, shuttle-run test and SES, favoring those included in the analysis (*p* < 0.05).

Information on the school context was obtained via a questionnaire developed by the authors in conjunction with the Vouzela city hall Education Department, namely: (i) school characterization (location; size and setting—rural, semi-urban and urban as determined by the Portuguese National Statistics Institute); (ii) policies and practices to stimulate physical activity and adequate nutrition; (iii) physical infrastructures of the school (playground, multi-sports roofed facilities and equipment for physical education); (iv) physical education (PE) classes; and (v) human resources (number of teachers in each school, academic degree of teachers, and teaching time). 

SES was identified using the school social support classification system (Portuguese Ministry of Education) [14]. Three SES groupings were identified: (i) level A: up to EUR 2.934 per year^−1^, children received books and school lunches; (ii) level B: from EUR 2.934 to EUR 5.896 per year^−1^, with half of level A support; (iii) level C: ≥EUR 5.897 per year^−1^ implies no support.

Height (cm) was measured, without shoes, using a portable stadiometer (Holtain, UK) holding the child’s head in the Frankfurt plane, (accuracy = 0.1 cm). Body mass (kg) was measured with a portable bioelectrical impedance scale (TANITA BC-418 MA Segmental Body Composition Analyzer; Tanita, Corporation, Japan) with children wearing light clothing and without shoes (precision = 0.1 kg). Body mass index (BMI) was calculated using the standard formula [BMI = body weight (kg)/height (m)^2^]. Birth weight (kg) was obtained retrospectively from children’s health booklets.

MSF (isometric strength and explosive strength/agility) [1] was assessed by: (i) Grip strength (HG in kg) via hand held dynamometry (Takei Physical Fitness Test GRIP-D, Japan). In all measurements the dynamometer was well fitted to the hand of the child, making the child as comfortable as possible. The most comfortable and stable positions were adopted to achieve optimal conditions for maximum muscle strength efforts. The child stood with the dynamometer beside their body, without touching the body. The orientation of each desired action was explained to the child while the investigator supported the movement. Verbal encouragement was given during the test. The dominant hand was previously established, as the children were evaluated with respect to their laterality [13]. We report the mean of the two trials performed with the dominant hand. The time set between the two trials was 2 min; (ii) Standing long jump (SLJ in cm) assessed lower body explosive strength via maximum jumping distance. The greatest distance from two trials was recorded; (iii) A shuttle run (SR in s) to measure explosive agility. Children ran from a starting line to a line of wooden blocks nine meters away, picked up a block, returned and placed the block on the starting line, before repeating the procedure. The quickest time of two trials, assessed by chronometry (Géonaute ON START 300), was recorded. The order of the testing was as follows: (i) anthropometric measurements, (ii) handgrip strength; (iii) SLJ and SH were assessed in either a multi-sports roofed or playground area of the schools. Tests were performed after a 10-min break between tests for the children to rest. All children participated in a warm-up period of 5 min, conducted by a member of the research team. 

The *Köperkoordinationstest für Kinder* (KTK) test battery [15] was used to assess GMC: (i) walking backwards—children walked backwards on three different balance beams, decreasing in width, three times. The number of successful steps was counted and recorded; (ii) hopping—children hop, with one leg, over an increasing number of foam squares, and successful completions were recorded (three attempts are allowed at each height); (ii) jumping sideways—with feet together, children jumped sideways over a wooden slat for 15 s. This was repeated twice, and the number of jumps was summed; (iv) moving sideways—children stood with both feet on one platform and had to place both hands on an adjacent platform, thus passing to the next platform. The total number of relocations over 20′s in two trials was recorded and summed. The sum of scores from four tests was used as a measure of total GMC (GMC_TS_) [15]. The KTK battery tests were arranged in the circuit manner in the order described above, with a 5-min break between each test.

PA was objectively measured using an Actigraph GT3X+ accelerometer (Actigraph, Pensacola, FL, USA) over 7 consecutive days (5 weekdays and 2 week-end days). Children were instructed to only remove the device during water activities and while sleeping at night. The accelerometer was placed on the iliac crest and held in place by an elastic belt with an adjustable clip. Each child received individual instructions and all parents received written instructions. ActiLife^®^ software 6.5.4 was used to process the recorded data immediately upon retrieval of each accelerometer. Children’s data, to be considered valid, had to have a minimum of at least 4 days (with at least one weekend day) and a minimum of 10 h of daily wear time. Any sequence of at least 20 consecutive minutes of zero activity counts was considered non-wear time [16]. Moderate-to-vigorous PA (MVPA) was derived using cut-points developed by Evenson et al. [17], defined as all activities greater than 574 counts/15 s epochs, and expressed in min∙day^−1^.

All measures were taken by trained team members of the Kinanthropometry and Motor Learning Labs (Faculty of Sport, University of Porto). Each year, a sample of ~60 children was randomly selected from each cohort and retested. Technical error of measurement (TEM) and ANOVA-based intraclass correlation (R) were used for reliability estimates: TEM = 0.17 cm in height, and 0.10 kg in weight; R-values ranged from 0.71 to 0.97 for PF tests, and from 0.78 to 0.91 in GMC tests. Finally, a systematic check was performed regarding errors in data entry and normality of variables’ distributions. 

Descriptive statistics by sex across age were calculated. A multilevel modeling approach was used [18]. As advocated [19], the time metric was anchored at 5 years of age (time 0), to ensure intercepts occurred within the data, such that age centers of 0, 1, 2, 3, 4, 5, and 6 years, correspond to 5, 6, 7, 8, 9, 10 and 11 years of age. All analyses were performed in SuperMix and model fit was simultaneously estimated by maximum likelihood [19]. Models were tested sequentially: Model 1 described the best developmental trajectory for each MSF test using age, age^2^, sex, age-by-sex and age-by-sex^2^ interactions as predictors; parameters were dropped from the final models if they were non-significant (*p* > 0.05); Model 2 included the cohort effect, non-varying (sex, birth weight, SES and school environment), and time-varying (BMI, GMC, MVPA) covariates. Individual models were developed for each of the three MSF markers (grip strength, standing long jump, and shuttle run). The significance level was set at *p* < 0.05.

## 3. Results

Boys’ and girls’ descriptive statistics (mean ± SD or percentages) are shown in Table 2; PF, BMI, and GMC_TS_ increased with increasing age. No differences in birth weights were found between sexes. Girls’ MVPA decreased until 9 years and increased thereafter; in boys this decrease was seen until 10 years and then increased. SES level C (≥ EUR 5.870 per year^−1^) was the most frequent category noted. 

The number of students per school varied, from a minimum of 5 to a maximum of 87 (Table 3). Most schools were located in rural areas and had policies and practices for PA. All schools had playgrounds without obstacles (any obstacle that could interfere with the children’s playground space) and their dimensions were generally greater than 70 m^2^. Only five schools had a sport center, three of them with more than 50 m^2^. All schools had equipment for PA; however, fifteen schools had only one infrastructure space for PA. The duration of PE classes ranged from 45 to 60 min and the time children spent active varied from 30 to 50 min. The same PE opportunities were offered to all children since schools employed PE specialists.

In model 1, no significant effects were identified for the interactions of age-by-sex or age^2^-by-sex, or for the school random effects. These factors were not included in the second model (Table 4). Regarding MSF tests, the averages for girls at 5 years (the centered age) were: HG = 7.74 kg, SLJ = 88.95 cm and SR = 15.49 s. Boys significantly outperformed girls at this age (+1.13 kg in HG, +7.88 cm in SLJ, and −0.33 s in SR) and this effect did not vary with age. The HG instantaneous velocity at 5 years was 1.43 ± 0.19 kg and the average acceleration (Age^2^) was also statistically significant (*β* = 0.06 ± 0.03, *p* < 0.05). A similar trend was observed for SLJ (velocity, *β* = 12.29 ± 1.20, *p* < 0.001; acceleration, *β* = 1.08 ± 0.17, *p* < 0.001), and SR (velocity, *β* = 0.93 ± 0.09 s, *p* < 0.001; acceleration, *β* = 0.07 ± 0.01, *p* < 0.001). Cohort effects were all significant (*p* < 0.05) in all MSF markers. Birth weight was only significantly related to SR, i.e., children born with higher birth weight demonstrated, on average, better agility during childhood (*β* = −0.18 ± 0.09, *p* < 0.05), i.e., faster SR test time. Further, SES was not significantly linked to any PF developmental trajectory. In contrast, children with higher BMI were stronger in HG (*β* = −0.35 ± 0.04, *p* < 0.001); less agile (*β* = 0.06 ± 0.01, *p* < 0.001), i.e., slower SR test time, and showed lower explosive strength (*β* = −0.93 ± 0.23, *p* < 0.001). Children with higher GMC_TS_ demonstrated better PF developmental trajectories (HG, *β* = 0.02 ± 0.003, *p* < 0.001; SLJ, *β* = 0.23 ± 0.02, *p* < 0.001; SR, *β* = −0.02 ± 0.001, *p* < 0.001). 

Additionally, more physically active children displayed a higher rate of increase in SLJ and SR, but not in the rate of HG development. Finally, children’s MSF developmental trajectories showed significant inter-individual differences, i.e., there was significant variation (*σ*^2^) in the longitudinal age-related trends across individuals (HG, *σ*^2^ = 0.21 ± 0.10, *p* < 0.05; SLJ, *σ*^2^ = 9.39 ± 3.42, *p* < 0.05; SR, *σ*^2^ = 0.06 ± 0.02, *p* < 0.001), meaning that children’s MSF developmental trajectories were significantly heterogeneous.

## 4. Discussion

The results show that isometric and explosive strength, as well as explosive agility, exhibit a curvilinear developmental trajectory (Age^2^ was significant) during childhood, which are analogous to other longitudinal studies [20]. When controlling for various covariates (birth weight, SES, BMI, GMC_TS_, and MVPA), boys outperformed girls in all PF developmental trajectories, corroborating other studies’ results [21]. Such sex-differences have been equated with biological and sociocultural aspects of development [2]. For example, it has been suggested that parents are more prone to let boys be involved in more intense rough play, whereas girls are treated as being more fragile [22] and participated more often and in more skill-based activities [23]. Boys also tend to demonstrate higher performance in GMC tasks [2] which is linked to the development of advanced neuromuscular function (e.g., inter- and intra-muscular coordination, motor unit recruitment/firing rate, proprioceptor sensitivity) that also promotes the development of strength and explosive strength/agility (i.e., musculoskeletal fitness) [24].

No relationship was found between children’s SES and PF trajectories, which is in contrast to other available data [12] that reported that school children attending schools with higher SES showed better profiles in aerobic capacity and BMI. This is probably related to the fact that Vouzela children, regardless of their SES inequality, generally have the same opportunities to develop their fundamental motor skills. Consequently, their MSF may not be related to SES because all children have mandatory physical education classes and equal access to city hall sporting facilities. 

BMI increases as children age, and the net expression of inter-individual differences as they increase in height and weight reflect factors linked to motor performance and PF levels [8]. We show that children’s changes in static strength and agility are positively related to increases in BMI; in contrast, SLJ is negatively associated. These results may be interpreted as follows: (i) with increasing age, children static strength increases and is related to improved neuromotor function and increases in absolute muscle mass [12]; (ii) agility development is linked to improved neuromotor function, GMC and more effective motor control, which all contribute to the effective acceleration and deceleration of the center of mass noted in repeated change of direction tasks, which was better expressed in children with lower BMI values; and (iii) although there is an assumption that vertical jumps are theoretically independent of body dimensions [25], and SLJ performance may be proportional to height [26], empirical data contradict these suggestions [25,26]. In contrast, force production is proportional to muscle physiological cross-sectional area to the power of 2, and body size is expected to be proportional to the power of 3, which means that greater body mass relative to height may impair the SLJ performance; thus, potentially explaining the negative relationship found between increases in BMI and decreased SLJ distance. 

Physiological mechanisms have previously linked birth weight with fat-free mass and muscle fiber development, and birth weight has been linked with physical performance [27]. In the present study we found no association of birth weight with static or explosive strength, or a positive relationship with agility. Robič Pikel et al. [28] found that although low birth weight children underperformed in explosive leg strength and speed tests, their agility was linked to GMC. Overall, the mechanisms of why specific relationships are found speaks to the importance of understanding both biological and experiential mechanisms for various aspects of physical performance and its development across time.

Children’s MVPA was not associated with HG strength but was positively interrelated to SLJ and SR performance. Although the longitudinal links between children’s PA and MSF trajectories are rarely investigated, Augste et al. [29] showed that PA affected not only children’s composite physical fitness baseline levels, but also their trajectory trends. Further, Breu et al. [30] indicated that children’s physical fitness was associated with their MVPA, and that they met the WHO guidelines at follow-up. Children’s participation in a variety of locomotor physical activities, specifically in early childhood, provides a logical link to tasks linked to lower extremity fitness and GMC performance tests. Finally, we speculate that since HG is specifically linked to static strength, which in turn is linked with size, it may not be as strongly linked to children’s general PA levels. 

It has been shown in primary school children that motor competence is positively related to physical fitness, and that change in motor competence development is linked to overweight or obesity [10]. We found that more coordinated children displayed better MSF developmental trajectories. Motor tasks in the GMC and MSF tests include multiplanar and multiaxial movements, and the combinations of different muscle contractions, integrated across different phases of movements, require a high degree of inter- and intramuscular coordination and control [24]. Similarly, the structure of explosive MSF tasks requires high levels of multijoin coordination and control inherent within GMC tasks.

Finally, we found that the school environment did not significantly explain differences in MSF development, probably because there was a high degree of similarity across schools. For example, the school-based curriculum in all Portuguese primary schools is applied in the same manner and rural areas have similar PA/sports facilities and physical education specialists. Further, school effects remain non-significant even when cohort effects were modelled and tested [31], although it is possible that children’s lives and educational histories within each age cohort could have impacted their MSF developmental trends.

In summary, our results illustrate the importance of understanding the relationships between various aspects of physical growth, SES and physical activity on MSF developmental during mid-childhood—an important time window that can be considered as a “sensitive” and transient period marked by a higher degree of plasticity in neuromuscular function. Our results may also increase physical educators’ and coaches’ awareness of these windows of opportunity for children’s MSF development within a life-long perspective. Even parents themselves need to always consider the links between physical growth and maturation that shape children’s MSF development. Ideally, it is important not only to focus on MSF outcomes, but to understand “how,” “why,” and “when” children are able, or not, to perform MSF tasks in terms of standardized movements and skills. In this way, such professionals and parents will be better equipped to provide appropriate learning environments, and conditions, linked to children’s intra- and inter-individual differences, allowing for more appropriate feedback that can impact children’s healthy trajectories.

A few limitations are acknowledged: (i) the GMDC-Vouzela study does not cover all Portuguese regions, and generalizability is questioned; (ii) a pure longitudinal study would have provided a more robust design. However, comprehensive testing of multiple intra-individual and environmental factors linked to physical development has never been conducted and the mixed-longitudinal approach is a highly viable and reliable design.

## 5. Conclusions

In conclusion, children’s MSF development exhibited a curvilinear trend with boys outperforming girls. The complex nature of interactions among MSF attributes, GMC and bodyweight status are difficult to untangle, specifically with the additional influence of biological growth and maturation variables. However, the unique nature of this investigation provides new novel insight on the overall physical development of children, supplying parents, physical education teachers and coaches with a more holistic view of children´s MSF development for future interventions.

## Figures and Tables

**Table 1 ijerph-20-02938-t001:** Number of children per cohort and age group.

Cohorts	Ages of Follow-Up								Total
**Cohort 1**	4	5	6						41
**Cohort 2**		5	6	7					65
**Cohort 3**			6	7	8				58
**Cohort 4**				7	8	9			59
**Cohort 5**					8	9	10		58
**Cohort 6**						9	10	11	67
									348

**Table 2 ijerph-20-02938-t002:** Descriptive statistics (mean ± SD or percentage) for girls (*n* = 177) and boys (*n* = 171) from 5 to 11 years.

**Tests**	**Girls**						
	**5 years** **(*n* = 28)**	**6 years** **(*n* = 58)**	**7 years** **(*n* = 73)**	**8 years** **(*n* = 73)**	**9 years** **(*n* = 86)**	**10 years** **(*n* = 47)**	**11 years** **(*n* = 23)**
**Physical fitness**							
Handgrip strength (kg^f^)	6.2 ± 1.1	8.0 ± 2.0	10.0 ± 2.5	12.7 ± 3.1	14.1 ± 2.9	16.4 ± 3.3	19.4 ± 3.5
Standing long jump (cm)	76.3 ± 16.3	93.7 ± 17.3	102.7 ± 16.5	113.8 ± 16.9	120.9 ± 18.2	131.3 ± 18.3	135.1 ± 17.6
Shuttle run (s)	16.6 ± 2.0	14.9 ± 1.5	14.0 ± 1.2	13.3 ± 1.1	12.9 ± 1.1	12.1 ± 1.0	11.8 ± 1.0
**Gestational Information**							
Birth weight (kg)	3.0 ± 0.6	3.0 ± 0.5	3.1 ± 0.5	3.2 ± 0.5	3.2 ± 0.5	3.2 ± 0.5	3.4 ± 0.4
**Anthropometry**							
BMI (kg·m^−2^)	16.7 ± 1.8	16.9 ± 2.0	17.7 ± 2.9	18.2 ± 3.2	19.0 ± 3.3	19.3 ± 3.4	20.2 ± 3.4
**Gross Motor Coordination**							
GMC_TS_ (points)	69.2 ± 22.5	103.0 ± 27.9	127.9 ± 28.8	155.8 ± 31.2	173.7 ± 32.4	207.1 ± 36.7	216.5 ± 42.7
**Physical activity**							
MVPA (min∙day^−1^)	64.6 ± 21.1	61.6 ± 18.9	60.3 ± 18.1	58.9 ± 19.3	54.0 ± 15.8	58.2 ± 17.8	64.2 ± 32.2
**Socioeconomic status**							
A (up to EUR 2.934 per year^−1^)	32.1%	15.5%	21.9%	20.5%	22.1%	21.3%	26.1%
B (EUR 2.934 to EUR 5.896 per year^−1^)	32.1%	36.2%	37.0%	26.0%	26.7%	21.3%	30.4%
C (≥EUR 5.870 per year^−1^)	35.7%	48.3%	41.1%	53.4%	51.2%	57.4%	43.5%
**Tests**	**Boys**						
	**5 years** **(*n* = 44)**	**6 years** **(*n* = 62)**	**7 years** **(*n* = 67)**	**8 years** **(*n* = 60)**	**9 years** **(*n* = 60)**	**10 years** **(*n* = 39)**	**11 years** **(*n* = 15)**
**Physical fitness**							
Handgrip strength (kg^f^)	7.0 ± 2.3	9.1 ± 2.4	11.5 ± 2.5	13.8 ± 3.4	15.5 ± 3.5	17.8 ± 4.4	20.1 ± 4.4
Standing long jump (cm)	85.6 ± 21.4	99.2 ± 19.2	114.7 ± 16.8	123.6 ± 18.6	130.4 ± 16.9	139.6 ± 15.3	143.5 ± 27.0
Shuttle run (s)	15.8 ± 2.1	14.7 ± 1.7	13.4 ± 1.1	13.0 ± 1.1	12.4 ± 1.1	11.7 ± 0.7	11.6 ± 1.1
**Gestational Information**							
Birth weight (kg)	3.3 ± 0.6	3.4 ± 0.5	3.4 ± 0.5	3.3 ± 0.5	3.2 ± 0.5	3.3 ± 0.6	3.4 ± 0.5
**Anthropometry**							
BMI (kg·m^−2^)	17.0 ± 1.9	17.0 ± 2.1	17.2 ± 2.3	17.9 ± 3.7	18.5 ± 3.9	18.8 ± 4.3	19.0 ± 3.1
**Gross Motor Coordination**							
GMC_TS_ (points)	66.7 ± 27.0	103.1 ± 33.4	129.9 ± 37.1	156.0 ± 33.2	174.6 ± 38.2	201.5 ± 28.3	215.5 ± 44.4
**Physical activity**							
MVPA (min∙day^−1^)	83.7 ± 23.5	82.6 ± 22.0	77.2 ± 24.1	74.8 ± 22.9	71.7 ± 20.8	68.5 ± 22.6	71.4 ± 19.8
**Socioeconomic status**							
A (up to EUR 2.934 per year^−1^)	15.9%	14.5%	14.9%	11.7%	15.0%	7.7%	-
B (EUR 2.934 to EUR 5.896 per year^−1^)	18.2%	24.2%	23.9%	25.0%	23.3%	33.3%	46.7%
C (≥EUR 5.870 per year^−1^)	65.9%	61.3%	61.2%	63.3%	61.7%	59.0%	53.3%

BMI = body mass index; GMC_TS_ = gross motor coordination score; MVPA = moderate-to-vigorous physical activity.

**Table 3 ijerph-20-02938-t003:** Descriptive statistics of the school environment.

Schools (*n* = 19)	Mean ± SD	Min–Max
**School Characterization**		
School size		
Number of children	23 ± 22	5–87
Number of teachers	4 ± 3	1–8
Ratio teachers/students	0.13 ± 0.11	0.05–0.55
	***n* (%)**	
School setting		
Rural	12 (63.2)	
Semi-urban	7 (36.8)	
**Policies and Practices for PA**		
Policies and practices	10 (52.6)	
Policies	5 (26.3)	
Practices	4 (21.1)	
**Physical Structure of the School**		
Playground		
Yes	19 (100)	
No	0 (0)	
Playground area		
With obstacles	19 (100)	
Without obstacles	0 (0)	
Playground dimension		
Smaller (10 m^2^ to 39 m^2^)	2 (10.5)	
Medium (40 m^2^ to 69 m^2^)	4 (21.1)	
Large (>70 m^2^)	13 (68.4)	
Multi-sports roofed		
Yes	5 (26.3)	
No	14 (73.7)	
Number of Infrastructures		
One Infrastructure	15 (78.9)	
Two Infrastructures	4 (21.1)	
Equipment for PA		
Yes	15 (78.9)	
No	4 (21.1)	
**PE Classes**		
Duration of PE classes		
45 min	6 (31.6)	
60 min	13 (68.4)	
Active in PE classes		
30 min	6 (31.6)	
40 min	4 (21.1)	
50 min	9 (47.3)	
**Human Resources**		
Academic Degree (all graduated)	19 (100)	

PA = physical activity; PE = physical education.

**Table 4 ijerph-20-02938-t004:** Parameter estimates (standard errors) for fixed and random effects of the three multilevel models.

	Handgrip Strength (kg^f^)	Standing Long Jump (cm)	Shuttle-Run Test (s)
**Regression coefficients (fixed effects—*β*)**			
Intercept (5 years)	7.74 (0.43) ***	88.95 (2.72) ***	15.49 (0.19) ***
Age (velocity)	1.43 (0.19) ***	12.29 (1.20) ***	−0.93 (0.09) ***
Age^2^ (acceleration)	0.06 (0.03) *	−1.08 (0.17) ***	0.07 (0.01) ***
Sex (boys)	1.13 (0.25) ***	7.88 (1.48) ***	−0.33 (0.09) ***
CE_c2_1	−0.80 (0.32) *	−8.81 (2.38) ***	0.18 (0.21) ^ns^
CE_c3_2	−0.87 (0.44) ^ns^	−12.48 (2.91) ***	0.62 (0.24) **
CE_c4_3	−1.18 (0.49) *	−8.18 (2.90) **	0.68 (0.22) **
CE_c5_4	−1.25 (0.46) **	−5.27 (2.52) *	0.62 (0.18) ***
CE_c6_5	−1.36 (0.35) ***	−4.35 (1.73) *	0.50 (0.11) ***
Birth weight (kg)	0.43 (0.23) ^ns^	0.25 (1.41) ^ns^	−0.18 (0.09) *
SES (level B)	−0.35 (0.35) ^ns^	−0.55 (2.13) ^ns^	−0.10 (0.13) ^ns^
SES (Level C)	−0.50 (0.32) ^ns^	−0.94 (1.93) ^ns^	0.02 (0.12) ^ns^
BMI (kg·m^−2^)	0.35 (0.04) ***	−0.93 (0.23) ***	0.06 (0.01) ***
GMC_TS_ (points)	0.02 (0.003) ***	0.23 (0.02) ***	−0.02 (0.001) ***
MVPA (min·day^−1^)	0.001 (0.003) ^ns^	0.08 (0.02) **	−0.003 (0.002) *
**Variance components (random effects—*σ*^2^)**			
Child Level Intercept	1.90 (0.58) **	217.67 (36.43) ***	2.01 (0.27) ***
Age	0.21 (0.10) *	9.36 (3.42) **	0.06 (0.02) ***
Covariance (intercept/age) Residual Level Intercept	0.09 (0.23) ^ns^ 1.95 (0.16) ***	−33.36 (10.79) ** 74.79 (6.25) ***	−0.34 (0.06) *** 0.44 (0.03) ***
**Model Summary**			
Number of estimated parameters	19	19	19

CE = cohort effects; CE_c2_1 is the overlapping effect of cohort 2 on cohort 1; SES = socioeconomic status; BMI = body mass index; GMC_TS_ = gross motor coordination score; MVPA = moderate-to-vigorous physical activity; ^ns^ = non-statistically significant; * *p* < 0.05; ** *p* < 0.01; *** *p* < 0.001.

## Data Availability

The data used in this study can be requested from one of the authors (J.M.) who is the curator of the broad data set of this study.

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
