# Peer review of "Modeling the Dynamics of Children’s Musculoskeletal Fitness"

_ijerph, 2023, doi:10.3390/ijerph20042938_

Round 1
Reviewer 1 Report
Thank you for sharing your work. Please see attached letter for comments.

Reviewer 2 Report
A review of the manuscript submitted to the IJERPH entitled “Modeling the dynamics of children’s musculoskeletal fitness”
The submitted paper considers a very significant topic which relates to the fitness development of children and the factors that influence this process. To justify this topic and to discuss the obtained results the authors used 28 references. Some of them are quite old and there are too many books cited, while more recent empirical studies should be included. The obtained data seems valuable as a cohort of 318 children were observed for a period of 3 years with evaluations of basic motor abilities as well as anthropometric variables and socio-economic status each year of the longitudinal study. The authors presented a hypothesis in which they suggest that MSF development is nonlinear and has greater trajectories in boys and these are influenced by physical activity. The methods section seems to describe the particular fitness tests adequately and the other measured variables. Physical activity was evaluated objectively with an ACTigraph. The anthropometric variables were also measured with proper tools and have good reliability. Trained personnel conducted the fitness tests and other measurements. Perhaps a little more detail should be given about the warm-up procedures, familiarity and the order of testing. Please use proper abbreviations for Si units such as (s) not sec. Also please change some of the key word as most of them appear in the title. Use the term variables and not parameters as in line 47. The results are presented in 4 tables which are rather long and not very informative. Consider presenting some of the results in a graphical form or mark the most significant results in bold. The main finding from the study is that strength and power variables exhibit a nonlinear development during childhood and boys reach better results in most physical fitness tests. The authors suggest that most sex differences are related to the type and intensity of physical activities taken up by boys. On the other hand the socio-economic status did not influence the physical fitness trajectories. This was explained by mandatory PE classes and access to school and city infrastructure. The research showed that children’s static strength and agility are positively related to BMI, while jumping ability is negatively related to increases in height and weight. This seems logical for increased body mass but explain in more detail why body height would have a negative effect on jumping ability. Overall no significant relationships were observed between birth weight and strength, while agility was related to GMC. The main finding of this longitudinal study indicates that both baseline level of physical fitness and its development is related to physical activity. The study also showed that GMC is positively related to physical fitness and showed that changes in motor competence development are linked to overweight and obesity. The limitations of the study are addressed, yet as mentioned before some of the details of testing were not presented. The English is quite good, yet a minor spell check is suggested, as several small mistakes occur, such as in line 287 where new novel should state novel. The paper can be accepted after a few minor corrections.
Reviewer 3 Report
The study aimed “to model children musculoskeletal fitness (MSF) developmental trajectories and identify individual differences related to effects of time-invariant, as well as time-varying covariates. 348 Portuguese children (177 girls) from six age-cohorts were followed for three years. MSF tests (handgrip strength, standing long jump and shuttle run), age, body mass index (BMI), socioeconomic status (SES), gross motor coordination (GMC) and physical activity (PA) were assessed.”
Comments and suggestions:
#1 Line 106: “MSF (isometric strength and explosive strength/agility) [1] was assessed by: (i) Grip strength (HG in kg) via hand held dynamometry (Takei Physical Fitness Test GRIP-D, Japan) using the dominant hand, recording maximum force over five seconds;”
Any prior calibration of this instrument?
Any information about reliability, accuracy and precision of this handgrip instrument?
Which measure was used? Maximal? Mean? Why not normalized to body weight?
Any familiarization period prior the experiments to avoid learning effect?
How about body, arms and hand position?
Any rest time between trials, or just one trial (I guess 2...)?
Was the dynamometer well fitted to the hand of the subjects?
#2 Line 137: “min∙day1.” ?
#3 Table 2: Unfortunately, I think all data, methods and results should be revised because there are some weird values, at least here in this table.
#4 Despite the quite huge sample size behind it (the authors should be congratulated for that…) and beyond the need to review methods and results, by reading the abstract, main document and conclusions, as far as I am aware about this research topic, i do not think this article brings any novelty.
Round 2
Reviewer 3 Report
Question 1: Line 106: “MSF (isometric strength and explosive strength/agility) [1] was assessed by: (i) Grip strength (HG in kg) via hand held dynamometry (Takei Physical Fitness Test GRIP-D, Japan) using the dominant hand, recording maximum force over five seconds;”
Any prior calibration of this instrument?
Any information about reliability, accuracy and precision of this handgrip instrument?
Which measure was used? Maximal? Mean? Why not normalized to body weight?
Any familiarization period prior the experiments to avoid learning effect?
How about body, arms and hand position?
Any rest time between trials, or just one trial (I guess 2...)?
Was the dynamometer well fitted to the hand of the subjects?
Authors answer: We would like to thank the reviewer very much for raising these issues. In fact, in all years of the project (and in other projects from which the instruments are used), the dynamometer was calibrated following the manual (Takei Scientific Instruments Co., Ltd.). In all measurements the dynamometer was well fitted to the hand of the child, making the child as comfortable as possible. The most comfortable and stable positions were adopted to achieve optimal conditions for maximum muscle strength efforts. The child stood with the dynamometer beside their body, without touching the body. The orientation of each desired action was explained to the child while the investigator supported the movement. Verbal encouragement was given during the test. The dominant hand was previously established, as the children were evaluated with respect to their laterality (more details see Reyes et al. 2018). We report the mean of the two trials performed with the dominant hand. The time set between the two trials was 2 minutes.
Regarding the body position during the handgrip test new information was added in the new draft.
Reference:
Reyes, A. C., Chaves, R., Baxter-Jones, A. D. G., Vasconcelos, O., Tani, G., & Maia, J. (2018). A mixed-longitudinal study of children's growth, motor development and cognition. Design, methods and baseline results on sex-differences. Annals of human biology, 45(5), 376–385. https://doi.org/10.1080/03014460.2018.1511828
Reviewer: Ok. Thus, please insert all these issues in the methods: “In all measurements the dynamometer was well fitted to the hand of the child, making the child as comfortable as possible. The most comfortable and stable positions were adopted to achieve optimal conditions for maximum muscle strength efforts. The child stood with the dynamometer beside their body, without touching the body. The orientation of each desired action was explained to the child while the investigator supported the movement. Verbal encouragement was given during the test. The dominant hand was previously established, as the children were evaluated with respect to their laterality (more details see Reyes et al. 2018). We report the mean of the two trials performed with the dominant hand. The time set between the two trials was 2 minutes.”
Question 3: Table 2: Unfortunately, I think all data, methods and results should be revised because there are some weird values, at least here in this table.
Authors answer: Although we appreciate the reviewer concern, we are uncertain which data, methods and results were weird. There was a “typo” in boys’ physical fitness where standing long jump and handgrip strength were wrongly placed (now corrected in the new draft) we would like to inform the reviewer that,
1. Three papers were already published with the data set from this research project in Annals of Human Biology (Reyes et al., 2018), Journal of Sport Sciences (Reyes et al., 2019), and Medicine and Science in Sports and Exercise (Pereira et al., 2022) and no reviewer found any problem with the data set.
2. We went again reading our methods section and found no problems other that adding more information as per reviewers’ suggestions.
3. Finally, we were left to know what is exactly weird in Table 2 results, apart from the “typo” we mentioned above and now corrected
References:
Reyes, A. C., Chaves, R., Baxter-Jones, A. D. G., Vasconcelos, O., Tani, G., & Maia, J. (2018). A mixed-longitudinal study of children's growth, motor development and cognition. Design, methods and baseline results on sex-differences. Annals of human biology, 45(5), 376–385. https://doi.org/10.1080/03014460.2018.1511828
Reyes, A. C., Chaves, R., Baxter-Jones, A. D. G., Vasconcelos, O., Barnett, L. M., Tani, G., Hedeker, D., & Maia, J. (2019). Modelling the dynamics of children's gross motor coordination. Journal of sports sciences, 37(19), 2243–2252. https://doi.org/10.1080/02640414.2019.1626570
Pereira, S., Reyes, A. C., Chaves, R., Santos, C., Vasconcelos, O., Tani, G. O., Katzmarzyk, P. T., Baxter-Jones, A., & Maia, J. (2022). Correlates of the Physical Activity Decline during Childhood. Medicine and science in sports and exercise, 54(12), 2129–2137. https://doi.org/10.1249/MSS.0000000000003013
Reviewer: Exactly, standing long jump and handgrip strength were wrongly placed.
Question 4: Despite the quite huge sample size behind it (the authors should be congratulated for that…) and beyond the need to review methods and results, by reading the abstract, main document and conclusions, as far as I am aware about this research topic, i do not think this article brings any novelty.
Authors answer: We thank the reviewer for her/his consideration about our work. Also, allow us to respectively disagree with the last comment.
- In our opinion, it is always a difficult task in precising the exact meaning of novelty in Sport Sciences research. We wonder if there is a list of indicators each of us should follow to find out if the work is really novel!
Reviewer: No.
- If such a list exists, but we are not aware of it, on how many indicators should we “tick” so that our work may be considered novel?
- For example, there are about 15-20 papers dealing with longitudinal data investigating gross motor coordination in children and adolescents. By what criteria should we say that these 15-20 papers are truly novel?
- In the field of Physical Fitness, and since 1950, there are hundreds of published papers, with the majority reporting cross-sectional data. Longitudinal data has only been reported in a few dozen. Are they all novel? Do they all tackle unforeseen matters? Is it possible that each paper is really unique in addressing original questions? After all, do we really have a kind of “tree of problems” when dealing with children physical fitness development?
- Finally, it is rare to find papers addressing MSF as we did with our study design, analytical techniques and sets of fixed and dynamic covariates.
Reviewer: Perhaps this reviewer was not clear in using the term "novelty" in the previous comment. In this case, which is (are) the relevant contribution (s) of your work to this “tree of problems” when dealing with children physical fitness development? Please highlight it in the abstract, discussion and conclusion section.
